# Utility and Limits of Lung Ultrasound in Childhood Pulmonary Tuberculosis: Lessons from a Case Series and Literature Review

**DOI:** 10.3390/jcm11195714

**Published:** 2022-09-27

**Authors:** Rosa Morello, Cristina De Rose, Vittoria Ferrari, Piero Valentini, Anna Maria Musolino, Daniele Guerino Biasucci, Luigi Vetrugno, Danilo Buonsenso

**Affiliations:** 1Department of Woman and Child Health and Public Health, Policlinic Gemelli Universitary Foundation, (IRCCS), Catholic University of Sacre Hearth, 00168 Rome, Italy; 2Department of Pediatric Emergency Medicine, Bambino Gesù Children’s Hospital, Istituto di Ricovero e Cura a Carattere Scientifico (IRCCS), 00165 Rome, Italy; 3Gruppo di Studio Pediatrico AdET; 4Intensive Care Unit, Department of Emergency, Anesthesiology and Intensive Care Medicine, Fondazione Policlinico Universitario Agostino Gemelli, Istituto di Ricovero e Cura a Carattere Scientifico (IRCCS), 00168 Rome, Italy; 5Department of Basic Biotechnological Sciences, Intensive and Perioperative Clinics, Catholic University of Sacre Hearth, 00168 Rome, Italy; 6Department of Medical, Oral and Biotechnological Sciences, University of Chieti-Pescara, 66100 Chieti, Italy; 7Global Health Research Institute, Institute of Hygiene, Catholic University of Sacre Hearth, 00168 Rome, Italy

**Keywords:** lung ultrasound (LUS), children, pulmonary tuberculosis, PTB, personalized medicine

## Abstract

Childhood pulmonary tuberculosis (PTB) diagnosis is often a challenge that requires a combination of history, clinical, radiological, immunological and microbiological findings. Radiological diagnosis is based today on the use of chest X-ray and chest CT that, in addition to being radio-invasive tools for children, are often not available in countries with low-resources. A non-invasive, easily usable and reproducible, low-cost diagnostic tool as LUS would therefore be useful to use to support the diagnosis of childhood PTB. Data on the use of LUS for the diagnosis and follow-up of childhood PTB are limited and in some respects contradictory. To help better define the potential role of LUS we have described the pros and cons of lung ultrasound method through a brief review of the studies in the literature and reporting some case series in which we describe clinical, laboratory, radiological results as well as detailed lung ultrasound findings of four children/adolescents with PTB.

## 1. Introduction

Childhood pulmonary tuberculosis (PTB) is increasingly recognized to contribute to the global tuberculosis (TB) burden. According to WHO’s 2019 TB Report, 7 million people were diagnosed and treated for TB in 2018–up from 6.4 million in 2017. Estimates of TB incidence and prevalence in children vary and have large uncertainty intervals, supporting the difficulty in establishing a diagnosis of TB in children [1,2]. Up to 50% of children clinically diagnosed with PTB test negative for Mycobacterium tuberculosis [1,2].

Timely identification of childhood TB is critical, as children are at risk for rapid progression and dissemination, especially if they are immunosuppressed, malnourished or young [1,2,3,4,5,6].

The PTB diagnosis in children is often a challenge that requires a combination of history, clinical, radiological, immunological and microbiological findings [1,2,3,4,5,6]. The radiological diagnosis is currently based on the use of chest-X-ray (CXR) and CT scans that are used to support a clinical diagnosis of childhood PTB. The leading radiographic finding in PTB in childhood is lymphadenopathy [1,7,8]. Parenchymal changes are also more frequent in the young age group and in the right lung [1,7,8]. Other lesions include pleuritis, atelectasis, destructive-cavitary lesions and miliary dissemination [1,7,8]. In resource-rich countries, CXR is usually the first investigation in a child with suspected PTB. However, radiographs are often inadequate for making the diagnosis of PTB because they have poor sensitivity for detecting mediastinal and hilar lymphadenopathy and have little specificity in characterizing parenchymal and pleural lesions [1,7,8,9]. CT remains the gold standard for the radiological diagnosis of PTB; it can define early features of TB (lymphadenopathy, nodules, small pleural effusion) before these become detectable in CXR [7,8,9,10]. However, CXR and especially CT cannot be used routinely and repeatedly in children for obvious reasons of radiation protection [11]. Moreover, they are expensive tools and their availability is limited in high TB-burden countries, low-resource settings, due to scarcity of both equipment and skilled radiological staff to perform and read the images [10,12].

In this context, a non-invasive, easily usable and reproducible, low-cost diagnostic tool as lung ultrasound (LUS) would therefore be useful to use to support the diagnosis of childhood PTB. In recent years, LUS has been increasingly used for the diagnosis of respiratory diseases in both adult and pediatric patients [13,14,15]. It is widely known that in children LUS has an excellent diagnostic accuracy for the diagnosis of pneumonia and other condition such as pneumothorax, respiratory distress syndrome (RDS), atelectasis and bronchiolitis [9,12,13,14,15]. LUS has also demonstrated its usefulness during the recent Coronavirus Disease 19 (COVID-19) pandemic [16]. Furthermore, due to the recent development of portable, low-cost ultrasound machines, chest ultrasound may become a cost-effective procedure especially in resource-limited settings [12].

Data on the use of LUS for the diagnosis and follow-up of childhood PTB are limited and in some respects contradictory [17,18,19,20,21,22,23,24,25,26,27]. Precisely because in some contexts such as in poor countries—which are also the countries in which childhood pulmonary tuberculosis is most widespread—LUS is the only tool that can be used, it is important to study its role in supporting the clinical diagnosis of childhood PTB. Therefore, for these reasons, we have described the pros and cons through a brief review of the studies in the literature and reporting some case series in which we describe clinical, laboratory, radiological results as well as detailed lung ultrasound findings of four children/adolescents with PTB.

Written informed consent was obtained from a parent or guardian before data collec-tion. The Institutional Review Board and Ethic Committee (prot. 36173/19 ID2729) ap-proved the study. All patients’ data were analyzed anonymously. The scans were conducted in the pediatric ward. LUS was performed by pediatricians with more than three years of experience in pediatric ultrasound and who perform it routinely in clinical practice. Ultrasound examinations were performed using a MyLab linear and curvilinear transducers and the small parts preset (Esaote SpA, Genoa, Italy). As for the “brief review of the literature”, we conducted a Pubmed search for “lung ultrasound and pulmonary tuberculosis” and reviewed all relevant articles.

## 2. Case Series

### 2.1. Case 1

In February 2021, a 17-year-old boy was transferred to the Pediatric Department of our hospital from a secondary care center for suspected PTB. He and his family had recently emigrated from Somalia and he resided in a reception center. A few months later, he developed cough, fever and respiratory failure. To confirm the diagnosis, the patient underwent radiological, immunological and microbiological evaluation. Anterior-posterior CXR showed extensive consolidation of the right upper lobe and the para-hilar field (Figure 1a). Chest CT characterized the parenchymal disease and in particular, it showed apical right upper lobe cavitation and ground glass areas on the right basal field (Figure 1b). Furthermore, it has documented lymphadenopathy in the right para-tracheal, pre-carinal, infra-carinal and right hilar regions. Interferon-gamma-release assay (IGRA) resulted positive; acid-fast bacilli were detected on microscopy (Ziehl-Neelsen stain) and the cultures were positive for Mycobacterium Tuberculosis. During hospitalization, we performed LUS that showed the absence of the pleural line, interrupted by a large consolidation with significant air content documented by thickened aerial broncograms, which extended into the right posterior mid-apical lung fields (Figure 1c).

### 2.2. Case 2

On June 2021, a 16-years-old girl came to our attention due to lumbar pain and fatigue. She came from Romania and her sister was diagnosed with tuberculosis ten years ago. CXR revealed left pleural effusion associated to an unspecific lung reduced transparency (Figure 2a). On the left lung fields, LUS showed a large pleural effusion with the pleural space with fibrin shoots associated with consensual lung atelectasis (Figure 2c). CT scans showed pleural empyema associated to atelectasis (Figure 2b). Besides, it showed multiple hilum-mediastinal lymphad-enopathies. The tuberculous disease was confirmed with microbiological tests.

### 2.3. Case 3

On October 2021, a 17-years-old boy referred to our ED for hemoptysis, fatigue, weight loss and night sweats. A month ago, he had emigrated from Somalia and he was accepted to a reception center. He was diagnosed with PTB in 2019 but he had poor compliance to treatment. A CXR documented reduced transparency in the upper right region and in the upper-middle left lung field (Figure 3a). CT showed—on the left lung: multiple cystic-like formations communicating with bronchial branches;—on the right lung: multiple air bubbles and many nodules with an appearance of ground glass. On the middle lobe, CT described “tree-in bud” pattern (Figure 3b). Based on history, clinical and radiological findings the patient started anti-tuberculosis regimen therapy. IGRA resulted positive, no evidence of acid-fast bacilli at microscopic examination and sputum cultures. During hospitalization, we performed LUS that showed: a pattern of interstitial syndrome; fragmented pleural line associated with many sub-pleural consolidations throughout the pulmonary field; a large consolidation, which extended into the left posterior mid-apical pulmonary field. (Figure 3c).

### 2.4. Case 4

On November 2021, a 16-years-old boy came to our attention at the pediatric outpatient clinic for clinical ultrasound to continue follow-up after initial diagnosis of pneumonia performed in the emergency room where he went for chest and abdominal pain. Chest-X-rays, performed at the ER observation, showed right basal pleural effusion associated with parenchymal inflammatory consolidation (Figure 4a). He was then diagnosed with pneumonia and discharged with broad-spectrum antibiotic therapy. At our follow-up evaluation, thoracic pain persisted. Further investigating the medical history, we also discovered that he was of Colombian origin and his mother was diagnosed with bacilliferous tuberculosis in 2016. We performed lung ultrasound that showed moderate pleural effusion predominantly anecogenic, but with some fibrin shoots inside. It extended into posterior-lateral and anterior right pulmonary field and was associated with consensual pulmonary atelectasis (Figure 4b). On suspicion of pulmonary TB, the patient was hospitalized. A Chest CT scan, microbiological, immunological examination were performed. A CT scans showed massive right pleural effusion (Figure 4c), disseminated micro-nodules and enlarged lymph nodes in pre-vascular, paratracheal, hilar and sub-carinal region. IGRA was positive, no evidence of acid-fast bacilli at microscopic examination of sputum.

## 3. Discussion

In the pediatric population, studies evaluating the potential role of LUS for pediatric tuberculosis as well as the potential for LUS in monitoring response to treatment are limited and the results are sometimes contradicting each other. In contrast, in the adult population, the role of LUS has been studied more systematically.

In particular, as regards the studies performed in the adult population, most showed a significant association between PTB diagnosis and some ultrasound findings of the lung parenchyma: multiple consolidations, apical consolidations, superior quadrant involvement and subpleural nodules [17,18,19]. Most of the studies suggest inserting LUS in the diagnostic algorithms of the PTB in adults, especially in low-income countries [17,18,19,20]. Furthermore, in adults, parenchymal lesions are wider and easier to be assessed by ultrasounds because they easily touch the pleura [21].

In adults, comparative studies among available radiological methods are limited and discordant in the results. According Fentress et al., LUS demonstrated poor ability to detect radiographically identified cavity. They concluded that LUS might have a role in screening and diagnosis of PTB only in areas without ready access to CXR [22]. Regarding pediatric studies, Heuvelings et al. for the first time they described lung ultrasound findings in children with suspected PTB. The authors found that, compared to adults, pleural effusion was more common, mediastinal lymph nodes were larger, resolution of consolidation occurred less commonly at 1-month follow-up and the proportional size reduction of a consolidation was lower. They suggested that LUS identified abnormalities suggestive of PTB with a high inter-reader agreement [23]. Subsequently, the same authors compared the pulmonary ultrasound findings with those detected on CXR in children hospitalized with suspected PTB. They demonstrated that LUS detected anomalies more frequently than CXR (LUS 72% vs. CXR 56%) [24].

Gaetano Rea et al. analyzed the potential role of LUS in in the diagnosis and management of patients with PTB. After comparing the limits and advantages of the method, they concluded that the use of LUS could have had a good rationale in circumstances that include avoiding exposure to ionizing radiation such as patients who need frequent follow-ups and in countries with limited resources. In fact, according to the authors there were still insufficient evidences to judge the diagnostic accuracy of LUS for a screening purpose in TB [25].

Bélard et al. described the utility of LUS in children with PTB; in particular, children with confirmed or unconfirmed PTB had a higher prevalence of POCUS findings than children with unlikely TB [18/58 (31%) and 36/119 (30%) vs. 8/55 (15%), *p* = 0.04 and *p* = 0.03, respectively]. Pleural effusion [*n* = 30 (13%)] or abdominal lymphadenopathy [*n* = 28 (12%)] was the most common findings and most of them resolved within 3 months of treatment. They concluded that POCUS can contribute to timely diagnosis of childhood PTB and to monitor treatment response [1].

Jacob Bigio et al. [26], in their review performed between January 2010 and June 2020, analyzed six scientific articles (five in adults and one in children, with a total sample size of 564 patients). They reported that studies had high risk of bias in many domains. In adults, subpleural nodule and lung consolidation were the lung ultrasound findings with the highest sensitivities, ranging from 72.5% to 100.0% and 46.7% to 80.4%, respectively. Only the study by Heuvelings et al. [24,25] provided detailed data on LUS findings in pediatric PTB (consolidations were the most common parenchymal alteration in children and LUS sensitivity was 45.6% in their detection). The authors concluded that there was no consensus on the optimal protocols for acquiring and analyzing LUS images for PTB. Therefore, they suggested that new studies, which minimized potential sources of bias, were necessary to assess the diagnostic accuracy of LUS in childhood PTB [26].

About our experience with third-level hospital resources, CXR is normally the first choice evaluation to support clinical diagnosis. However, being a third-level center and having the possibilities and resources, due to the low sensitivity and specificity of the radiography [1,7,8] CT scans are performed in most cases. In particular, in all four cases CT scans showed its superiority in identifying and characterizing parenchymal alterations, the pleural effusion and the lymphadenopathy as already highlighted by the literature.

Analyzing the role of LUS in our case series, it has been very useful in the diagnostic process of PTB for several reasons:It showed in cases 1 and 3 unusual pleural abnormalities (Figure 1c and Figure 3c) that might correspond with the subpleural lesions described in the CT;It identified in cases 1 and 3 large consolidations (Figure 1c and Figure 3c) with significant air content documented by thickened aerial broncograms. We have noticed that the latter have different characteristics compared to the air bronchograms present in bacterial or viral pneumonia [9]. These large consolidations could represent the tubercular cavitation seen at chest CT, but all these findings still need to be studied to better clarify their meaning;In the cases 2 and 4, it demonstrated its superiority compared to CXR in characterizing the pleural effusion and consolidation of an atelectatic nature (Figure 2c and Figure 4c); LUS is therefore useful as a non-invasive diagnostic instrumental support to identify and characterize a picture of tuberculous pleuritis.

However, the limitations of LUS were also evident in our case series:
It was not possible to evaluate the hilar lymph nodal disease (Figure 1b), the hallmark of pediatric PTB, because of the interposition of air between the ultrasound and hilar lymph nodes; andIt was not possible to detect lesions to far from the pleura; in adults, parenchymal lesions are wider and easier to be assessed by ultrasounds [8].

## 4. Conclusions

LUS is an emergent diagnostic tool with clear advantages (Table 1) such as it is free of ionizing radiation, fast, repeatable at low cost and easy to learn. Due to the paucity and contradiction of available data in childhood PTB, no conclusions can be drawn on diagnostic accuracy and you need to be aware of LUS limitations (Table 1). Many other studies are needed in the pediatric population to clarify the significance of the LUS findings found in childhood PTB and therefore to standardize the role of the LUS in the diagnosis of childhood PTB.

Meanwhile, based on currently available data, on the case series shown by us and on studies performed in extremely resource-limited setting [12,27,28], we believe LUS can be useful—in high-resources countries for the follow-up of TB lung consolidations which touch the pleura and—as instrumental diagnostic support in particular conditions such as in low-resource countries with high childhood mortality PTB due to missed or delayed diagnosis [1,6,12,27,28].

## Figures and Tables

**Figure 1 jcm-11-05714-f001:**
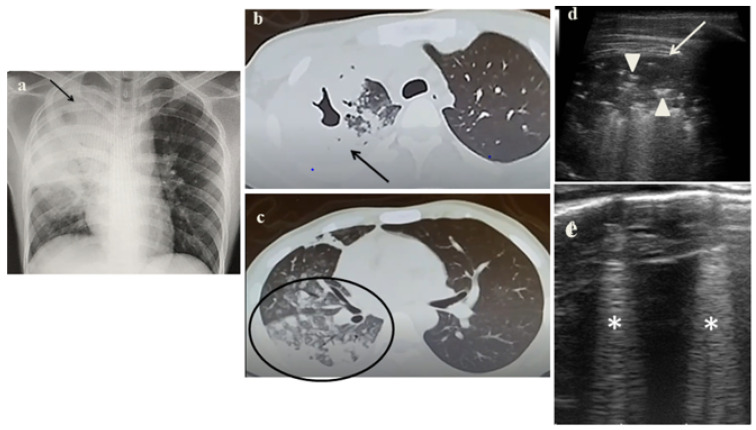
(**a**) Chest radiography shows—on the right lung apex—an extensive area of reduced transparency with cavitated air bubble of about 2-cm. (**b**,**c**) A representative axial section from thoracic computed tomography (CT) scan reveals a right lung apex cavitation (*arrow*) and ground glass areas (*black circle*) on the entire dependent lung. (**d**,**e**) Grayscale lung ultrasound examination shows—on the right posterior mid-apical lung—the absence of the pleural line, interrupted by a large consolidation (*arrow*) in which there is a significant air content documented by thickened aerial broncograms (*arrowhead*);—on the right posterior and lateral basal lung—long confluent vertical art—ifacts (*asteristic*).

**Figure 2 jcm-11-05714-f002:**
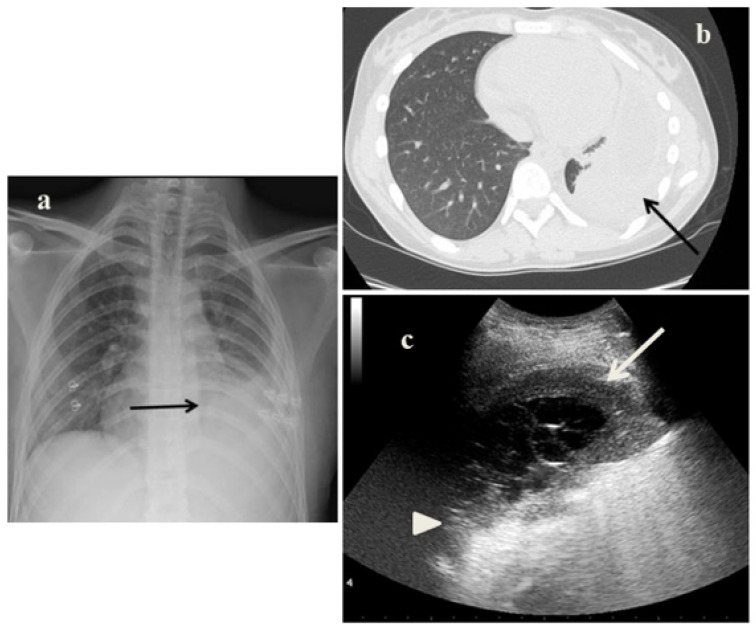
(**a**) Chest radiography shows left pleural effusion associated to an unspecific lung opacity (*arrow*). (**b**) A representative axial section from thoracic computed tomography (CT) scan reveals—on the left mid-basal lung—pleural empyema associated to lung atelectasis (*arrow*). (**c**) Grayscale lung ultrasound examination shows—on the left mid-basal lung, a large pleural effusion with the pleural space filled with fibrin shoots (*arrow*) associated with consensual lung atelectasis (*arrowhead*).

**Figure 3 jcm-11-05714-f003:**
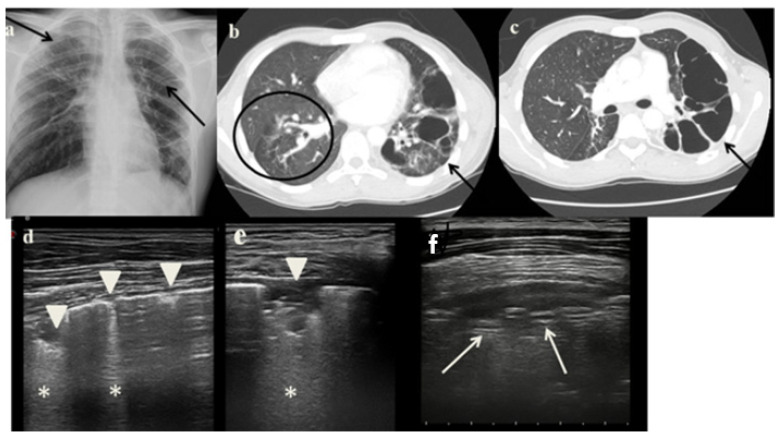
(**a**) Chest radiography shows a lung ill-defined pulmonary opacity on the upper right lung and on the upper-middle left lung. (**b**,**c**) A representative axial section from thoracic computed tomography (CT) scan reveals—on the left lung, multiple cystic-like formations communicating with bronchial branches (*arrow*);—on the right lung, multiple air bubbles and many nodules with an appearance of ground glass and a “tree-in bud” pattern (*arrowed*). (**d**–**f**) Grayscale lung ultrasound examination shows—on the right lung: a pattern of long confluent artifacts (*asterisks*) and numerous interruptions of the pleural line associated with many sub-pleural consolidations throughout the lung field (*arrowhead*);—on the left lung: a large consolidation in which there is a significant air content documented by thickened aerial broncograms (*arrow*).

**Figure 4 jcm-11-05714-f004:**
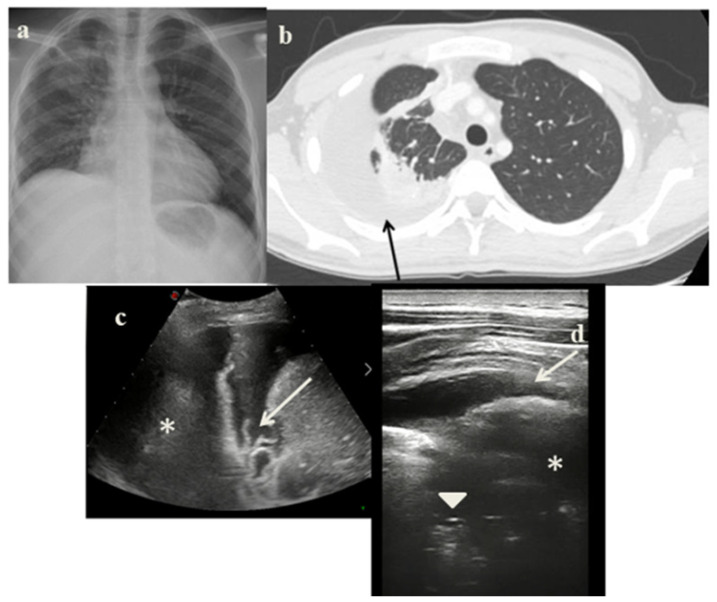
(**a**) Chest radiography shows right basal pleural effusion associated to a lung ill-defined pulmonary opacity. (**b**) A representative axial section from thoracic computed tomography (CT) scan reveals massive right pleural effusion (*arrow*). (**c**,**d**) Grayscale lung ultrasound examination shows—on the right mid-basal lung, a large pleural effusion predominantly anecogenic but with some fibrin shoots inside (*arrow*) associated with consensual pulmonary atelectasis (*asterisks*) characterized by static and parallel air bronchograms (*arrowhead*).

**Table 1 jcm-11-05714-t001:** Summary of the pros and cons of the use of LUS in pulmonary TB in pediatric age on the basis of studies in the literature and on the basis of our experience.

UTILITY	LIMITS
Clinical diagnosis feasible in association with clinical-anamnestic and epidemiological data Easily detect and characterize pleural effusions during the tuberculous pleurisy Easily detect and characterize the alterations of the pleural line Easily detect and characterize of subpleural consolidations by defining the inflammatory or atelectasis nature Monitoring and Follow-up feasible and cheap, allow also to monitor complete resolution of pleural and subpleural lesions Free from radiation Repeatable Availability at bed-side (new pocket devices can be used with mobile phones) Cheap (bed side devices and wireless probe more diffused, easily available and cheaper, rechargeable with sunlight) A few hours training is sufficient to learn to detect TB subpleural ultrasound lesions	Clinical diagnosis feasible only in association with clinical and epidemiological data Inaccurate and specific clinical diagnosis Non-specific diagnosis for tuberculous cavity lesions It identifies and characterizes pleural and subpleruic lesions only Some of the ultrasound lesions found in pulmonary TB still need to be systematically studied to understand their significance Tuberculous lesions that do not touch the pleura cannot be detected and monitored May be very difficult to detect mediastinal lynphnodes, partic-ularly if they are too deep to be reached by the ultrasound

## Data Availability

Available upon request to the corresponding author.

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
