# Peer review of "Utility and Limits of Lung Ultrasound in Childhood Pulmonary Tuberculosis: Lessons from a Case Series and Literature Review"

_jcm, 2022, doi:10.3390/jcm11195714_

Round 1

Reviewer 1 Report

The reviewed paper is devoted to the possibilities and limitations of the application of lung ultrasound in diagnosing tuberculosis in children. The Authors have undertaken a very difficult task since the relevant literature is relatively scarce.

The four case studies of pediatric patients with tuberculosis presented in this paper are very interesting. I am, however, wondering about the descriptions of ultrasound images concerning patient 3 and patient 4 – I would like to ask the Authors to respond to my comments.

Case 3

Lines 154-159

The first lung ultrasound (LUS) image presents small subpleural consolidations with visible vertical C-line artifacts (asterisks). This is not the pattern of the interstitial syndrome, and definitely not the ‘white lung’. The situation is similar as regards the second LUS image provided in the description of this case.

Descriptions of the images need to be corrected.

I suggest changing the image taken over the left lung because it is not consistent with the presented CT image.

Case 4

Lines 180-183

The figure lacks letters that would mark specific elements in particular images. 

The described fibrin (arrow) is not correctly marked in the presented images.

Additionally, I suggest symbols that mark particular elements in LUS images be white – they will be definitely more visible and clear for readers.

Introduction and Discussion sections contain too many paragraphs. In my opinion, the sentences should be grouped into longer sections.

Literature: items 8 and 22 feature the same publication.

Additional questions to the Authors: Did the described patients undergo ultrasound monitoring during treatment or was the ultrasound examination performed only once? If they were monitored – what was the evolution of the pathologies?

Author Response

Dear Editor,

we went through the comments of the anonymous Reviewers, that we sincerely thank. We have made several changes, accordingly to the suggestions of the Reviewers. The quality of the manuscript has been definitely improved thanks to the comments.

The revisions made now we have tracked them in the manuscript using the "Track Changes" function in Microsoft Word.

We hope that now the paper is suitable for publication in JCM.

Sincerely,

Cristina De Rose

   (on the behalf of co-authors)

REPLY TO REFEREE

Reviewer's Comments:

Reviewer #1

The reviewed paper is devoted to the possibilities and limitations of the application of lung ultrasound in diagnosing tuberculosis in children. The Authors have undertaken a very difficult task since the relevant literature is relatively scarce.

The four case studies of paediatric patients with tuberculosis presented in this paper are very interesting. I am, however, wondering about the descriptions of ultrasound images concerning patient 3 and patient 4 – I would like to ask the Authors to respond to my comments.

  • We thank the Reviewer very much for appreciating our manuscript and for understanding the aim. Following his suggestions, we think that our work has overall improved with images and concepts that are more understandable for each reader. We have answered point by point as follows.

-Case 3

Lines 154-159

The first lung ultrasound (LUS) image presents small subpleural consolidations with visible vertical C-line artifacts (asterisks). This is not the pattern of the interstitial syndrome, and definitely not the ‘white lung’. The situation is similar as regards the second LUS image provided in the description of this case.

Descriptions of the images need to be corrected.

  • We have changed the description of the pictures as follows: " Chest radiography shows an lung ill-defined pulmonary opacity on the upper right lung and on the upper-middle left lung. b and c. A representative axial section from thoracic computed tomography (CT) scan reveals - on the left lung, multiple cystic-like formations communicating with bronchial branches (arrow); -on the right lung, multiple air bubbles and many nodules with an appearance of ground glass and a “tree-in bud” pattern (arrowed). d,f and g. Grayscale lung ultrasound examination shows - on the right lung: a pattern of long confluent artefacts (asterisks) and numerous interruptions of the pleural line associated with many sub-pleural consolidations throughout the lung field (arrowhead); - on the left lung: a large consolidation in which there is a significant air content documented by thickened aerial broncograms (arrow)”.

I suggest changing the image taken over the left lung because it is not consistent with the presented CT image.

  • From the ultrasound point of view, what we see is what we have shown in the figures (d,f and g). To date it is not possible to make specific comparisons between the different methods because studies with this purpose have not yet been carried out on a fairly large population of patients. However, what we want to show is that ultrasound detects (in correspondence of cavity lesions touching the pleura detected on CT) consolidations inside which there is air represented by thick air bronchograms.

-Case 4

Lines 180-183

The figure lacks letters that would mark specific elements in particular images.

The described fibrin (arrow) is not correctly marked in the presented images.

Additionally, I suggest symbols that mark particular elements in LUS images be white – they will be definitely more visible and clear for readers.

 è We corrected the caption as follows: "a. Chest radiography shows right basal pleural effusion associated to a lung ill-defined pulmonary opacity. B. A representative axial section from thoracic computed tomography (CT) scan reveals massive right pleural effusion (arrow). c and d. Grayscale lung ultrasound examination shows - on the right mid-basal lung, a large pleural effusion predominantly anecogenic but with some fibrin shoots inside (arrow) associated with consensual pulmonary atelectasis (asterisks) characterized by static and parallel air bronchograms (arrowhead) ".

è We modified the image both by adding letters to each figure and by inserting the white arrows on the ultrasound images

-Introduction and Discussion sections contain too many paragraphs. In my opinion, the sentences should be grouped into longer sections.

à Thanks for the hint, we have grouped sentences related to the same topic into longer paragraphs as suggested.

-Literature: items 8 and 22 feature the same publication

à We have mistakenly entered the wrong reference therefore we have modified by inserting the right one at number 22.

-Additional questions to the Authors: Did the described patients undergo ultrasound monitoring during treatment or was the ultrasound examination performed only once? If they were monitored – what was the evolution of the pathologies?

  • We followed the patients even at a distance with a different timing for each patient depending on the type of lesion they presented in the acute phase. In fact, the ultrasound follow-up timing has not yet been defined in a standardized way either by us or by other authors. However, we found in all of them a slow resolution of the ultrasound lesions with the persistence of outcomes characterized by small consolidations and mostly single long vertical artifacts that were not clinically significant.

Reviewer 2 Report

Thanks very much for the opportunity to review this manuscript. I commend the authors for their efforts in preparing this manuscript which I enjoyed reading.

I am writing my comments from the perspective of a diagnostic radiologist. I do not encounter many cases of tuberculosis in my practice. I would assume my experience is similar to many radiologists and clinicians around the world who do not work in areas where tuberculosis is prevalent. Therefore, I find the topic discussed in this paper of particular use to clinicians who may not have much personal clinical experience with tuberculosis for when they encounter a patient where tuberculosis is a differential consideration.

In addition, this paper is particularly useful as a demonstration of the applications of lung ultrasound. Although lung ultrasound is gaining more attention as clinicians recognize its utility, many still are unaware of its excellent diagnostic capabilities. For full disclosure, given my lack of personal clinical experiences with pulmonary tuberculosis, my review primarily focuses on the imaging aspects of your paper. 

Abstract:

-Appropriate setting of the context and scope of the paper.

Introduction:

-“The radiological diagnosis is currently based on the use of Chest-X ray (CXR) and CT 43 scans that are used to support a clinical diagnosis of childhood PTB. Leading radiographic 44 finding in PTB in childhood is lymphadenopathy (1,7,8). Parenchymal changes are also 45 more frequent in the young age group and in the right lung (1,7,8). Other lesions include 46 pleuritis, atelectasis, destructive-cavitary lesions and miliary dissemination (1,7,8)”

Do you know the prevalence of these findings? It would be helpful for me as a radiologist to know the percentage of patients presenting with lymphadenopathy, consolidation, cavitary lesions, etc. to have a better idea of how this disease presents.

-I think your introduction provides the background on this topic in an effective way describing pulmonary tuberculosis and the potential for lung ultrasound to be an effective tool in its diagnosis. Do you know of any statistics regarding the number of people infected with tuberculosis worldwide? That may add more clarity to the scope of the problem for the reader especially for those of us who do not practice in these settings where tuberculosis is prevalent.

-For your “brief review of the literature,” are you able to provide information on how your search was conducted? For example, did you search for lung ultrasound and pulmonary tuberculosis in PubMed and review all of the articles? Can you please elaborate as to how your literature search was conducted? Without this information some readers may question possible bias or exclusion of relevant studies.

-You state, “Ultrasound examinations were performed using a MyLab linear transducer at 12 87 MHz and the small parts preset (EsaoteSpA, Genoa, Italy).”

-Some of the images use a curvilinear probe. I would recommend revising this sentence to reflect that.

Case Series:

I think your figures will require major revisions. Do you have a radiologist author? It would be helpful to consult with a radiologist if you have not done this.

-I have numerous concerns about figure 1.

The figure parts are not broken up appropriately. In my opinion, the figures all need to be labeled with a different letter. The different sections of the CT scan and different ultrasound images cannot have the same letter. This is because you need to distinguish the findings on each image. For example, you say there are ground glass opacities on the CT in 1b, but there are two figure 1bs.

What is the arrow on the chest x-ray? This needs to be specified in the legend what you are arrowing. If you intended to have this labeling a cavitary lesion, it does not correlate to the cavitary lesion on the chest CT which is at the lung apex.

You say that there are coronal CT images, but these are axial CT images.

You arrowhead groundglass opacity but the entire dependent lung demonstrates ground glass opacity. I would recommend consideration to circling the entire area of groundglass opacity or adding multiple arrows.

This is a bit picky, but there are no anatomic lung “fields.” I personally would recommend not using this term although I do understand what you are saying and I know people use this in common vernacular. The ground glass opacity you are describing in figure 1b is in the dependent lung, not the basilar lung. Basilar lung would be near the diaphragm.

In figure 1b, there is no lymphadenopathy able to be visualized on a lung window. To visualize lymphadenopathy, you would need to provide a soft tissue window. I would recommend obtaining a soft tissue window and arrowing the lymphadenopathy for the readers who do not have a radiology background.

Your lung ultrasound images do show a nice example of consolidation with air bronchograms. However, both of these images are nearly identical and having two does not really add anything from my perspective. Do you have any additional lung ultrasound images of either 1) the normal side for comparison or 2) B-lines corresponding to the area of ground glass opacity.   

-I have a concerns regarding figure 2

I’m not sure what a pulmonary reduced transparency is? Please adjust the wording. In addition, again findings on the chest x-ray need to be arrowed and explained. You have an arrow but do not explain what this is arrowing.

Again, this is not a coronal CT image. This is an axial CT image.

What are you arrowing on the CT image? The arrow looks like it is not pointing anything out to me.

You are not really showing the empyema on the CT scan.

On the lung ultrasound image, this is a picture of empyema. However, you have a “black on black” arrow which is a suboptimal choice. The * is not showing atelectasis. This is either artifact or air.

I’ve published a paper in this journal before so I understand the template can be a bit difficult to adjust your images. I do not think the way the images look appears professional. I would recommend adjusting all the figure images so they are in line and/or as close to the same size as possible.

I have concerns regarding figure 3:

Again, these all should be individually lettered.

I have never heard the term “reduced transparency” do you mean “ill-defined pulmonary opacity”

I don’t think you are describing where the pulmonary opacity is effectively on the chest x-ray.

These are axial CT scans not coronal scans.

You put an arrowhead but do not note what this is describing

You have not shown the communication with the bronchial branches well if they are communicating with the branches

What is the significance of the split pleural line? I have never head

In regards to your ultrasound findings, I think the point for this case should be more emphasized the lung ultrasound is not effective for evaluating cavitary lesions. Lung ultrasound does not penetrate air therefore making it unsuitable for evaluation of cavitary lesions. This point should be made clear.

I question the consolidation in C based on the CT findings and x-ray. Are you sure this is a real consolidation?

-I have similar concerns about figure 4. Please consult with a radiologist. I do not see any effusion in the chest x-ray. The CT again is axial. These figures are not lettered. Some of the image annotations are black on black. Are you sure this is atelectasis and not consolidation?

Discussion:

-In this paper you have a few short paragraphs which I leave to your discretion as authors. I typically would try to make my paragraphs at least 3-4 sentences.  I would recommend not having a single sentence paragraph though which is present in your discussion.

-Much of the information in this discussion reads to me as you are summarizing articles without synthesizing the information. For example, you say what each paper found but you don’t really make conclusions or interpret the different findings.

-Your reference 24 I believe is a pre-print and not published. I have personally never heard or experienced the use of lung ultrasound for pulmonary nodule evaluation. I have done a brief literature search to recheck and believe that my assessment is correct and that these pulmonary nodules are not amenable to ultrasound examination. The physics of lung ultrasound are such that in an aerated lung, ultrasound does not penetrate the tissues. Miliary nodules are also so small that they may be outside the resolution of lung ultrasound. Regardless to avoid this altogether, I might recommend removing it altogether from the paper.

-I cannot find what you are referencing in your reference 30. Please recheck this. Also, I’m not sure how this relates to pediatric tuberculosis. In general, I would like more emphasis on the differences between adult and pediatric tuberculosis.

-Also, of note, your patients are at the upper limits of what is considered a pediatric patient. I do not have enough clinical knowledge on tuberculosis to comment on the pathological differences between adult and pediatric tuberculosis. Nonetheless, I wonder if these patients closer to adulthood would not have classic pediatric tuberculosis.

-I would adjust your description of your cases after a thorough review by a radiologist to fix the errors in your figures.

-I think much of this information could be reorganized into tables. You could have a table for the pros and cons of lung ultrasound. You could also have a table where you describe the literature you are citing. The text of the discussion can reference these articles and synthesize the information for the audience. As it is now, your discussion to me reads more of a “findings dump” in which you state findings without explaining them or contextualizing them. The main question of the role of lung ultrasound in evaluation of pulmonary tuberculosis I do not believe has been answered sufficiently.

From my perspective, we know lung ultrasound is excellent for evaluating consolidation, pleural effusion, and interstitial lung disease. In pediatric pulmonary TB, the question is are the patients presenting with these findings? Lung ultrasound as you state does not evaluate for lymphadenopathy. It is also not good at evaluating cavitary lesions as I explained above.   

-Are your cases representative of the clinical spectrum of pediatric pulmonary tuberculosis as described in the literature? It is unclear to me if these are typical cases or atypical cases? Did you capture the entire spectrum of imaging findings?

-This paper is multidisciplinary in nature involving pulmonology, internal medicine, pediatrics, and radiology among other specialties. As with such a broad audience, I think it is important to explain basic concepts that members of one specialty might not be as familiar with. For example, even describing the basics of lung ultrasound including the basic physics and sensitivity and specificity for pneumonia and effusion would be helpful. Similarly, for those of us with less clinical background, explaining more about the clinical manifestations, treatment, and prognosis of TB would be helpful. This doesn’t need to be included in this discussion but could be scattered throughout the paper in the introduction as well.

I am a native English speaker and also have a few English comments that you may find helpful:

Abstract:

“However, a non-invasive, 20 easily usable and reproducible, low-cost diagnostic tool as LUS would therefore be useful to use to 21 support the diagnosis of childhood PTB especially in countries with low resources and therefore be 22 used as a clinical tool, which can support in the clinical diagnosis, disease severity determination 23 and treatment response monitoring.”

-This is a run-on sentence.

Introduction:

“The radiological diagnosis is currently based on the use of Chest-X ray (CXR) and CT scans that are used to support a clinical diagnosis of childhood PTB”

-I do not typically see chest x-ray capitalized.

“Leading radiographic finding in PTB in childhood is lymphadenopathy (1,7,8).”

-This is a sentence fragment. It should read, “The leading radiographic…

“In this context, a non-invasive, easily usable and reproducible, low-cost diagnostic 60 and accurate tool would therefore be useful to use to support the diagnosis of childhood 61 PTB particularly in high burden - low resource settings in which access to other radiolog- 62 ical methods is limited and where often only ultrasound is available as the only diagnostic 63 tool.”

-This is a run-on sentence.

“The main setting was represented by pediatric ward.“

-I would recommend, “The scans were conducted in the pediatric ward.” If this accurate. It may also be helpful to elaborate if this was a PICU or what type of hospital the scans were obtained in. Were these hospitalized inpatients?

Case Series:

“On February 2021, a 17-years-old boy was transferred to the Pediatric Department of 91 our hospital from a secondary care center for suspected PTB”

-This should read, “In February 2021, a 17-year-old boy…”

“To confirm diagnosis, the patient un- 94 derwent radiological, immunological and microbiological evaluation”

-This should read, “To confirm the diagnosis…”

“CXR revealed left pleural effusion associated to an unspecific pulmonary reduced 118 transparency (Figure 2a). On the left lung fields, LUS showed a large pleural effusion with 119 the pleural space filled with fibrin shoots associated with consensual pulmonary atelecta- 120 sis (Figure 2c). CT scans confirmed the ultrasound picture: it showed pleural empyema 121 associated to atelectasis (Figure 2b). Besides, it showed multiple hilum-mediastinal lym- 122 phadenopathies. The tuberculous disease was subsequently confirmed with microbiolog- 123 ical tests. In addition to initiating anti tuberculosis treatment, the patient underwent thor- 124 acotomy surgery to remove empyema and to perform pulmonary decortication to facili- 125 tate re-expansion of the lung itself.”

-There are multiple English errors in this paragraph. Some of which I am not sure what you are trying to convey. I would recommend re-editing this section.

Conclusion:

“LUS is an emergent and attractive technique with clear advantages such as radiation- 269 free, fast, repeatable, low cost, and easy-to-learn. “

-This is grammatically incorrect.

Author Response

Dear Editor,

we went through the comments of the anonymous Reviewers, that we sincerely thank. We have made several changes, accordingly to the suggestions of the Reviewers. The quality of the manuscript has been definitely improved thanks to the comments.

The revisions made now we have tracked them in the manuscript using the "Track Changes" function in Microsoft Word.

We hope that now the paper is suitable for publication in JCM.

Sincerely,

Cristina De Rose

   (on the behalf of co-authors)

Reviewer #2

Thanks very much for the opportunity to review this manuscript. I commend the authors for their efforts in preparing this manuscript which I enjoyed reading.

I am writing my comments from the perspective of a diagnostic radiologist. I do not encounter many cases of tuberculosis in my practice. I would assume my experience is similar to many radiologists and clinicians around the world who do not work in areas where tuberculosis is prevalent. Therefore, I find the topic discussed in this paper of particular use to clinicians who may not have much personal clinical experience with tuberculosis for when they encounter a patient where tuberculosis is a differential consideration.

In addition, this paper is particularly useful as a demonstration of the applications of lung ultrasound. Although lung ultrasound is gaining more attention as clinicians recognize its utility, many still are unaware of its excellent diagnostic capabilities. For full disclosure, given my lack of personal clinical experiences with pulmonary tuberculosis, my review primarily focuses on the imaging aspects of your paper. 

  • We thank the Reviewer very much for appreciating our manuscript and for understanding the aim. We also thank the Reviewer for all the suggestions he has given us especially regarding the imaging aspects of our work. We hope that the images with the related captions have improved. In any case, following the suggestions of the reviewer, we think that our work has overall improved with images and concepts that are more understandable for each reader. We have answered point by point as follows.

 Abstract:

-Appropriate setting of the context and scope of the paper.

Introduction:

-“The radiological diagnosis is currently based on the use of Chest-X ray (CXR) and CT 43 scans that are used to support a clinical diagnosis of childhood PTB. Leading radiographic 44 finding in PTB in childhood is lymphadenopathy (1,7,8). Parenchymal changes are also 45 more frequent in the young age group and in the right lung (1,7,8). Other lesions include 46 pleuritis, atelectasis, destructive-cavitary lesions and miliary dissemination (1,7,8)”

Do you know the prevalence of these findings? It would be helpful for me as a radiologist to know the percentage of patients presenting with lymphadenopathy, consolidation, cavitary lesions, etc. to have a better idea of how this disease presents.

  • We have not specified the presentation percentages of each individual condition because in reality all lesions are part of the pattern of possible presentation of pulmonary TB and for some of them the frequency with which they occur is not yet known because no studies have been done to know it. However we have inserted the references where it is possible to find some of this information.

-I think your introduction provides the background on this topic in an effective way describing pulmonary tuberculosis and the potential for lung ultrasound to be an effective tool in its diagnosis. Do you know of any statistics regarding the number of people infected with tuberculosis worldwide? That may add more clarity to the scope of the problem for the reader especially for those of us who do not practice in these settings where tuberculosis is prevalent.

-->According to the WHO's 2019 TB Report, 7 million people were diagnosed and treated for TB in 2018 - up from 6.4 million in 2017. We added the concept to the text as suggested.

-For your “brief review of the literature,” are you able to provide information on how your search was conducted? For example, did you search for lung ultrasound and pulmonary tuberculosis in PubMed and review all of the articles? Can you please elaborate as to how your literature search was conducted? Without this information some readers may question possible bias or exclusion of relevant studies.

--> As for the "short literature review", we conducted a Pubmed search for "pulmonary ultrasound and pulmonary tuberculosis" and reviewed all relevant articles. We have added the concept into the text as suggested.

-You state, “Ultrasound examinations were performed using a MyLab linear transducer at 12 87 MHz and the small parts preset (EsaoteSpA, Genoa, Italy).”

-Some of the images use a curvilinear probe. I would recommend revising this sentence to reflect that.

--> We have corrected the paragraph because we have actually also used the curvilinear probe in patients where the subcutaneous tissue was more present.

Case Series:

I think your figures will require major revisions. Do you have a radiologist author? It would be helpful to consult with a radiologist if you have not done this.

-I have numerous concerns about figure 1.

-->We have modified the caption related to figure 1 as follows: “a. Chest radiography shows - on the right lung apex - an extensive area of reduced transparency with cavitated air bubble of about 2-cm. b and c. A representative axial section from thoracic computed tomography (CT) scan reveals a right lung apex cavitation (arrow) and ground glass areas (black circle) on the entire dependent lung. d and f. Grayscale lung ultrasound examination shows - on the right posterior mid-apical lung - the absence of the pleural line, interrupted by a large consolidation (arrow) in which there is a significant air content documented by thickened aerial broncograms (arrowhead); - on the right posterior and lateral basal lung - long confluent vertical artifacts (asteristic)”.

The figure parts are not broken up appropriately. In my opinion, the figures all need to be labeled with a different letter. The different sections of the CT scan and different ultrasound images cannot have the same letter. This is because you need to distinguish the findings on each image. For example, you say there are ground glass opacities on the CT in 1b, but there are two figure 1bs.

  • We have labeled the figures with different letters as suggested.

What is the arrow on the chest x-ray? This needs to be specified in the legend what you are arrowing. If you intended to have this labeling a cavitary lesion, it does not correlate to the cavitary lesion on the chest CT which is at the lung apex.

  • Yes, the intention was to label the cavitary lesion at the apex of the lung, so we moved the arrow.

You say that there are coronal CT images, but these are axial CT images.

You arrowhead groundglass opacity but the entire dependent lung demonstrates ground glass opacity. I would recommend consideration to circling the entire area of groundglass opacity or adding multiple arrows.

  • Sorry for the mistake regarding the CT images section, we corrected.

--> We have indicated the entire area of ground glass opacity of the entire dependent lung with a black circle as suggested.

This is a bit picky, but there are no anatomic lung “fields.” I personally would recommend not using this term although I do understand what you are saying and I know people use this in common vernacular. The ground glass opacity you are describing in figure 1b is in the dependent lung, not the basilar lung. Basilar lung would be near the diaphragm.

  • We modified it as suggested.

Your lung ultrasound images do show a nice example of consolidation with air bronchograms. However, both of these images are nearly identical and having two does not really add anything from my perspective. Do you have any additional lung ultrasound images of either 1) the normal side for comparison or 2) B-lines corresponding to the area of ground glass opacity.   

  • We agree with the reviewer on the usefulness of showing a different ultrasound image and in particular we have inserted as suggested an image showing long confluent vertical artifacts corresponding to the area of ground glass opacity of the dependent right lung touching the pleura.

-I have a concerns regarding figure 2

--> We have modified the caption related to figure 1 as follows: “a. Chest radiography shows left pleural effusion associated to an unspecific lung opacity (arrow). b. A representative axial section from thoracic computed tomography (CT) scan reveals - on the left mid-basal lung - pleural empyema associated to lung atelectasis (arrow). c. Grayscale lung ultrasound examination shows - on the left mid-basal lung, a large pleural effusion with the pleural space filled with fibrin shoots (arrow) associated with consensual lung atelectasis (arrowhead)”.

I’m not sure what a pulmonary reduced transparency is? Please adjust the wording. In addition, again findings on the chest x-ray need to be arrowed and explained. You have an arrow but do not explain what this is arrowing.

--> As for the captions of the radiographs, we decided to modify the text by literally copying the report that the radiologist produced when the radiograph was made. We have added the arrow indication.

Again, this is not a coronal CT image. This is an axial CT image.

What are you arrowing on the CT image? The arrow looks like it is not pointing anything out to me.

You are not really showing the empyema on the CT scan.

  • We have once again corrected the error regarding the type of scans.

--> We changed the CT image in order to better show the lesion described.

On the lung ultrasound image, this is a picture of empyema. However, you have a “black on black” arrow which is a suboptimal choice. The * is not showing atelectasis. This is either artifact or air.

  • We have inserted the white markers and replaced the * with the arrowhead indicating the static air bronchograms of consensual lung atelectasis.

I’ve published a paper in this journal before so I understand the template can be a bit difficult to adjust your images. I do not think the way the images look appears professional. I would recommend adjusting all the figure images so they are in line and/or as close to the same size as possible.

  • We have changed the size of all the figures so that they are as the same size as possible.

I have concerns regarding figure 3:

  • We have changed the description of the pictures as follows: " a. Chest radiography shows a lung ill-defined pulmonary opacity on the upper right lung and on the upper-middle left lung. b and c. A representative axial section from thoracic computed tomography (CT) scan reveals - on the left lung, multiple cystic-like formations communicating with bronchial branches (arrow); -on the right lung, multiple air bubbles and many nodules with an appearance of ground glass and a “tree-in bud” pattern (arrowed). d,f and g. Grayscale lung ultrasound examination shows - on the right lung: a pattern of long confluent artefacts (asterisks) and numerous interruptions of the pleural line associated with many sub-pleural consolidations throughout the lung field (arrowhead); - on the left lung: a large consolidation in which there is a significant air content documented by thickened aerial broncograms (arrow)”.

Again, these all should be individually lettered.

--> We have once again marked each figure with a different letter.

I have never heard the term “reduced transparency” do you mean “ill-defined pulmonary opacity”

I don’t think you are describing where the pulmonary opacity is effectively on the chest x-ray.

  • We modified as suggested using the term "ill-defined pulmonary opacity" and indicated by arrows the lesions on the figure.

These are axial CT scans not coronal scans.

  • We have once again corrected and modified.

You put an arrowhead but do not note what this is describing

You have not shown the communication with the bronchial branches well if they are communicating with the branches

  • We changed the annotation on the figure.

What is the significance of the split pleural line? I have never head

In regards to your ultrasound findings, I think the point for this case should be more emphasized the lung ultrasound is not effective for evaluating cavitary lesions. Lung ultrasound does not penetrate air therefore making it unsuitable for evaluation of cavitary lesions. This point should be made clear.

I question the consolidation in C based on the CT findings and x-ray. Are you sure this is a real consolidation?

--> It is described that the pleural line, in addition to being thinned and / or thickened, can be irregular with interruptions due to the presence of consolidations of different sizes or long and / or shorts artifacts (Musolino AM, Supino MC, Buonsenso D, et al. Roman Lung Ultrasound Study Team for Pediatric COVID-19 (ROMULUS COVID Team). Lung Ultrasound in Children with COVID-19: Preliminary Findings. Ultrasound Med Biol. 2020 Aug; 46 (8): 2094-2098).

-->As far as cavity lesions are concerned, to date it is not possible to make specific comparisons between the different methods because studies with this regard have not yet been carried out on a fairly large population of patients. However, what we want to show is that ultrasound detects (in correspondence of cavity lesions touching the pleura detected on CT) consolidations inside which there is air represented by thick aerial bronchograms.

-I have similar concerns about figure 4. Please consult with a radiologist. I do not see any effusion in the chest x-ray. The CT again is axial. These figures are not lettered. Some of the image annotations are black on black. Are you sure this is atelectasis and not consolidation?

  • We corrected the caption as follows: "a. Chest radiography shows right basal pleural effusion associated to a lung ill-defined pulmonary opacity. B. A representative axial section from thoracic computed tomography (CT) scan reveals massive right pleural effusion (arrow). c and d. Grayscale lung ultrasound examination shows - on the right mid-basal lung, a large pleural effusion predominantly anecogenic but with some fibrin shoots inside (arrow) associated with consensual pulmonary atelectasis (asterisks) characterized by static and parallel air bronchograms (arrowhead) ".
  • The caption referring to the radiography was taken from the report produced by the radiologist.
  • We modified the image both by adding letters to each figure and by inserting the white arrows on the ultrasound images
  • As for consolidation on ultrasound, it has the characteristics of atelectasis because it is characterized by static, fine and parallel airborne bronchograms (Musolino AM, Supino MC, Buonsenso D, et al. Roman Lung Ultrasound Study Team for Pediatric COVID -19 (ROMULUS COVID Team). Lung Ultrasound in Children with COVID-19: Preliminary Findings. Ultrasound Med Biol. 2020 Aug; 46 (8): 2094-2098).

Discussion:

-In this paper you have a few short paragraphs which I leave to your discretion as authors. I typically would try to make my paragraphs at least 3-4 sentences.  I would recommend not having a single sentence paragraph though which is present in your discussion.

-->Thanks for the suggestion, we have modified the discussion by grouping the shorter sentences and related to the same topic in a single paragraph

-Much of the information in this discussion reads to me as you are summarizing articles without synthesizing the information. For example, you say what each paper found but you don’t really make conclusions or interpret the different findings.

-->Our proposal was to report the conclusions drawn by the authors of the various studies and finally draw our conclusions based also on our experience.

-Your reference 24 I believe is a pre-print and not published. I have personally never heard or experienced the use of lung ultrasound for pulmonary nodule evaluation. I have done a brief literature search to recheck and believe that my assessment is correct and that these pulmonary nodules are not amenable to ultrasound examination. The physics of lung ultrasound are such that in an aerated lung, ultrasound does not penetrate the tissues. Miliary nodules are also so small that they may be outside the resolution of lung ultrasound. Regardless to avoid this altogether, I might recommend removing it altogether from the paper.

-->We agree with the reviewer, therefore as suggested, we have eliminated from the discussion the paragraph to which the reference itself refers because it can actually be a confounding factor for the reader.           

-I cannot find what you are referencing in your reference 30. Please recheck this. Also, I’m not sure how this relates to pediatric tuberculosis. In general, I would like more emphasis on the differences between adult and pediatric tuberculosis.

-->The insertion of reference number 30 was an error relating to the initial draft of the manuscript. Therefore we have eliminated the aforementioned reference.

-Also, of note, your patients are at the upper limits of what is considered a pediatric patient. I do not have enough clinical knowledge on tuberculosis to comment on the pathological differences between adult and pediatric tuberculosis. Nonetheless, I wonder if these patients closer to adulthood would not have classic pediatric tuberculosis.

-->Usually, children oldern than 10 years of age have a so called "adult type TB", characterized mostly by cavernaes or large consolidations, rather than mediastinal TB or miliary TB.

-I think much of this information could be reorganized into tables. You could have a table for the pros and cons of lung ultrasound. You could also have a table where you describe the literature you are citing. The text of the discussion can reference these articles and synthesize the information for the audience. As it is now, your discussion to me reads more of a “findings dump” in which you state findings without explaining them or contextualizing them. The main question of the role of lung ultrasound in evaluation of pulmonary tuberculosis I do not believe has been answered sufficiently.

-->As suggested by the reviewer, we have created a table (Table 1 in the text) summarizing the pros and cons of the use of LUS in pediatric pulmonary TB based on our experience and the studies reported. We have modified the conclusions as follows: “LUS is an emergent diagnostic tool with clear advantages (Table 1) such as it is free of ionizing radiation, fast, repeatable at low cost and easy to learn. Due to the paucity and contradiction of available data in childhood PTB, no conclusions can be drawn on diagnostic accuracy and you need to be aware of LUS limitations (Table 1). Many other studies are needed in the pediatric population to clarify the significance of the LUS findings found in childhood PTB and therefore to standardize the role of the LUS in the diagnosis of childhood PTB. Meanwhile, based on currently available data, on the case series shown by us and on studies performed in extremely resource-limited setting (13,28,29), we believe LUS can be useful - in high-resources countries for the follow-up of TB lung consolidations which touch the pleura and - as instrumental diagnostic support in particular conditions such as in low-resource countries with high childhood mortality PTB due to missed or delayed diagnosis (1,6,13,28,29).”

-From my perspective, we know lung ultrasound is excellent for evaluating consolidation, pleural effusion, and interstitial lung disease. In pediatric pulmonary TB, the question is are the patients presenting with these findings? Lung ultrasound as you state does not evaluate for lymphadenopathy. It is also not good at evaluating cavitary lesions as I explained above.   Are your cases representative of the clinical spectrum of pediatric pulmonary tuberculosis as described in the literature? It is unclear to me if these are typical cases or atypical cases? Did you capture the entire spectrum of imaging findings?

-->The lung lesions shown by both radiographic and ultrasound images are typical of the pediatric TB spectrum. Among our cases, however, the miliary form is missing, which also occurs in the pediatric age.

-This paper is multidisciplinary in nature involving pulmonology, internal medicine, pediatrics, and radiology among other specialties. As with such a broad audience, I think it is important to explain basic concepts that members of one specialty might not be as familiar with. For example, even describing the basics of lung ultrasound including the basic physics and sensitivity and specificity for pneumonia and effusion would be helpful. Similarly, for those of us with less clinical background, explaining more about the clinical manifestations, treatment, and prognosis of TB would be helpful. This doesn’t need to be included in this discussion but could be scattered throughout the paper in the introduction as well.

--> TB in children can range from asymptomatic or mild subtle disease to life threatening disease such as miliary TB or large pulmonary consolidations with bronco-lymphnodal fistulae. LUS has been proved to be highligh sensitive and specific for recognizing lung consolidations and pleural effusions (Musolino AM, Tomà P, De Rose C, Pitaro E, Boccuzzi E, De Santis R, Morello R, Supino MC, Villani A, Valentini P, Buonsenso D. Ten Years of Pediatric Lung Ultrasound: A Narrative Review. Front Physiol. 2022 Jan 6;12:721951. doi: 10.3389/fphys.2021.721951. PMID: 35069230; PMCID: PMC8770918.) 

I am a native English speaker and also have a few English comments that you may find helpful:

..>We thank the reviewer for the valuable linguistic suggestions, they were very useful in improving the style of the article and being clearer for all readers.

Abstract:

“However, a non-invasive, 20 easily usable and reproducible, low-cost diagnostic tool as LUS would therefore be useful to use to 21 support the diagnosis of childhood PTB especially in countries with low resources and therefore be 22 used as a clinical tool, which can support in the clinical diagnosis, disease severity determination 23 and treatment response monitoring.” This is a run-on sentence.

--> We changed the sentence as follows “Radiological diagnosis is based today on the use of chest X-ray and chest CT that, in addition to being radio-invasive tools for children, are often not available in countries with low-resources. A non-invasive, easily usable and reproducible, low-cost diagnostic tool as LUS would therefore be useful to use to support the diagnosis of childhood PTB.”

Introduction:

“The radiological diagnosis is currently based on the use of Chest-X ray (CXR) and CT scans that are used to support a clinical diagnosis of childhood PTB”

-I do not typically see chest x-ray capitalized. 

 -->  We changed it to lowercase as suggested

“Leading radiographic finding in PTB in childhood is lymphadenopathy (1,7,8).”

-This is a sentence fragment. It should read, “The leading radiographic…    

--> We changed the sentence as follows: “The leading radiographic finding in PTB in childhood is lymphadenopathy”

“In this context, a non-invasive, easily usable and reproducible, low-cost diagnostic 60 and accurate tool would therefore be useful to use to support the diagnosis of childhood 61 PTB particularly in high burden - low resource settings in which access to other radiolog- 62 ical methods is limited and where often only ultrasound is available as the only diagnostic 63 tool.”

-This is a run-on sentence.

 --> We changed the sentence as follows “In this context, a non-invasive, easily usable and reproducible, low-cost diagnostic tool as lung ultrasound (LUS) would therefore be useful to use to support the diagnosis of childhood PTB”.

“The main setting was represented by pediatric ward.“

-I would recommend, “The scans were conducted in the pediatric ward.” If this accurate. It may also be helpful to elaborate if this was a PICU or what type of hospital the scans were obtained in. Were these hospitalized inpatients?

            --> We changed it as suggested  

Case Series:

“On February 2021, a 17-years-old boy was transferred to the Pediatric Department of 91 our hospital from a secondary care center for suspected PTB”

-This should read, “In February 2021, a 17-year-old boy…”

--> We changed it as suggested  

“To confirm diagnosis, the patient un- 94 derwent radiological, immunological and microbiological evaluation”

-This should read, “To confirm the diagnosis…”

--> We changed it as suggested

“CXR revealed left pleural effusion associated to an unspecific pulmonary reduced 118 transparency (Figure 2a). On the left lung fields, LUS showed a large pleural effusion with 119 the pleural space filled with fibrin shoots associated with consensual pulmonary atelecta- 120 sis (Figure 2c). CT scans confirmed the ultrasound picture: it showed pleural empyema 121 associated to atelectasis (Figure 2b). Besides, it showed multiple hilum-mediastinal lym- 122 phadenopathies. The tuberculous disease was subsequently confirmed with microbiolog- 123 ical tests. In addition to initiating anti tuberculosis treatment, the patient underwent thor- 124 acotomy surgery to remove empyema and to perform pulmonary decortication to facili- 125 tate re-expansion of the lung itself.”

-There are multiple English errors in this paragraph. Some of which I am not sure what you are trying to convey. I would recommend re-editing this section.

--> We changed this paragraph as follows: “CXR revealed left pleural effusion associated to an unspecific lung reduced transparency (Figure 2a). On the left lung fields, LUS showed a large pleural effusion with the pleural space with fibrin shoots associated with consensual lung atelectasis (Figure 2c). CT scans showed pleural empyema associated to atelectasis (Figure 2b). Besides, it showed multiple hilum-mediastinal lymphad-enopathies. The tuberculous disease was confirmed with microbiological tests”.

Conclusion:

“LUS is an emergent and attractive technique with clear advantages such as radiation- 269 free, fast, repeatable, low cost, and easy-to-learn. “

-This is grammatically incorrect.

  • We changed the sentence as follows: “LUS is an emergent diagnostic tool with clear advantages such as it is free of ionizing radiation, fast, repeatable at low cost and easy to learn.”

Round 2

Reviewer 1 Report

Thank you for responding to suggestions and corrections.